# A New Perspective on Robot Ethics through Investigating Human–Robot Interactions with Older Adults



Anouk van Maris [1,]*[ID], Nancy Zook [2], Sanja Dogramadzi [3], Matthew Studley [1], Alan Winfield [1] and Praminda Caleb-Solly [4][ID]

1  Bristol Robotics Laboratory, Faculty of Environment and Technology, University of the West of England, Bristol BS16 1QY, UK; Matthew2.Studley@uwe.ac.uk (M.S.); alan.winfield@brl.ac.uk (A.W.)
2  Faculty of Health and Applied Sciences, University of the West of England, Bristol BS16 1QY, UK; nancy.zook@uwe.ac.uk
3  Department of Automatic Control & Systems Engineering, University of Sheffield, Sheffield S10 2TN, UK; s.dogramadzi@sheffield.ac.uk
4  School of Computer Science, University of Nottingham, Nottingham NG7 2RD, UK; praminda.caleb-solly@nottingham.ac.uk
*  Correspondence: anouk.vanmaris@brl.ac.uk

**Abstract:** This work explored the use of human–robot interaction research to investigate robot ethics. A longitudinal human–robot interaction study was conducted with self-reported healthy older adults to determine whether expression of artificial emotions by a social robot could result in emotional deception and emotional attachment. The findings from this study have highlighted that currently there appears to be no adequate tools, or the means, to determine the ethical impact and concerns ensuing from long-term interactions between social robots and older adults. This raises the question whether we should continue the fundamental development of social robots if we cannot determine their potential negative impact and whether we should shift our focus to the development of human–robot interaction assessment tools that provide more objective measures of ethical impact.

**Keywords:** social robot; older adults; ethics; deception; attachment

## 1. Introduction

The growth in robot applications has led to human–robot interactions (HRI) in many different areas such as industry [1] and healthcare [2]. These HRI applications can take place in the form of physical interactions (e.g., exoskeleton support [3]) or social interactions (e.g., health support, customer service [4]). Social robots can be defined as robots that employ social interactions to meet the needs of its human users [5]. They can be used for applications such as companionship [6] and as a therapeutic device [7]. The goal of these robots is to improve people's quality of life [8]. An important factor in reaching this goal is the robot's ability to interact more reliably, intuitively and naturally with people. Expressing artificial emotions during interactions can be an asset, as it has been found to increase the use of the robot, resulting in people being able to fully benefit from the services that the robot provides [9]. The expression of artificial emotion can also build empathy and induce trust [10], which is a factor in the acceptance of social robots [11].

Interactions with robots reach beyond the scope of human–robot interactions, for example, animal–robot interactions (e.g., [12–15]). This work focuses specifically on older adults as they are particularly likely to benefit from social robots [16] due to the increase in loneliness [17] and health issues [18] as people live longer. However, the likelihood of negative consequences during human–robot interactions (HRI) may also be greater for older adults due to age-related impairments that might result in a higher level of vulnerability [19,20].

Ethical concerns have been expressed in robot ethics literature, not only focusing on the impact of social robots but human–robot collaborations in general. Example concerns

include the loss of privacy, loss of control, infantilisation, attribution of responsibility, loss of human contact, emotional deception, and emotional attachment [19,21–26]. Furthermore, these concerns not only apply to the people directly interacting with the robot but also to others that are engaged in improving the quality of life for these people [27,28].

Many principles and frameworks have been published to address ethical concerns in robotics and Artificial Intelligence [29]. However, even though these concerns have been raised in the literature, they are often not investigated and analysed further as part of real-world trials and studies [30]. Due to the lack of real-world studies in this area, it should not be assumed that all the obstacles and ethical concerns are known [28]. We aim to address this by investigating ethical concerns of social robots that express artificial emotions through human–robot interaction experiments.

A long-term human–robot interaction study was conducted with older adults to investigate the occurrence and impact of emotional deception and emotional attachment. In this study, participants' experiences when interacting with a robot that displayed artificial emotions was compared with their experiences when interacting with a robot that did not display emotions. This study provided rich insights on user experiences, highlighting potential negative consequences of expression of artificial emotions by a social robot during human–robot interactions. This study has been described in prior work [31]; thus, an overview will be presented here to provide context for the insights made regarding the use of HRI research to investigate robot ethics. The main contributions from this paper consist in using the findings from this research to further investigate robot ethics and raise the fundamental question of whether we should continue to develop these robots if we cannot sufficiently investigate their impact or if we should shift the focus of our research to include the development of HRI assessment tools that can help us do so.

The next section provides a background on emotional deception and emotional attachment and discusses why they are ethical concerns. This is followed by a description of the methods and materials used to conduct the study with older adults. The fourth section provides the results of this study, followed by a discussion of the insights following these results. Finally, the last section of the paper describes the conclusions of the research.

## 2. Emotional Deception and Emotional Attachment

The ethical concerns investigated in this work are emotional deception and emotional attachment. These two concerns were selected given the lack of knowledge in real-world applications in HRI as to the potential negative impact of expression of artificial emotion by social robots on users.

### 2.1. Emotional Deception

A person is being deceived when they receive false information in order to benefit the person who provides the information [32]. If an agent gives a false impression of having an emotional state, this is known as emotional deception. in the context of robotic deception, three forms of robotic deception have been identified [33]: external state deception, superficial state deception, and hidden state deception. If a robot misrepresents the impression of having an emotional state, this is known as superficial state deception. The complication of robot deception compared to philosophical deception is that robot deception is often unintentional [24,33]. The display of artificial emotions is often used to ensure a more natural interaction [34] and not with the goal to deceive a person. It is argued that such deception is 'harmless fun' [24] and should not be regarded as deception [33]. However, it should be determined that emotional deception is indeed harmless through user experience before it is disregarded as an ethical concern.

### 2.2. Emotional Attachment

Attachment generally refers to the feelings of connections one has for other people. These attachment connections can be extended to pets [35] and objects [36]. As people tend to react socially towards computers, also known as the Media Equation [37], it is not

hard to imagine that people can become attached to robots as well. This attachment will likely become more intense as robots become more advanced [38], suggesting a robot that displays artificial emotion may be more likely to evoke higher levels of attachment than those without emotion. Attachment to a robot can be beneficial as it may lead to increased use of the robot [9], but it may also have negative consequences such as increasing a person's level of dependence on the robot [19] and social isolation [39,40]. Therefore, user experiences following attachment should be acquired to ensure the benefits of eliciting attachment through the expression of artificial emotions outweigh the potential harm.

## 3. Materials and Methods

### 3.1. Experimental Procedure

A mixed between–within subject design was used for the experiment. The duration of the experiment was six weeks. In the first week, the experimenter would inform the participant about the experiment. Weeks two to five consisted of two sessions per week—a total of eight sessions—where participants interacted with the robot. In week six, participants were debriefed by the experimenter and a final interview was conducted.

Some participants would only interact with a robot that did not express artificial emotions during the interaction. For other participants, the robot would express artificial emotions for four sessions and would not do so for the other four sessions—this order was counterbalanced. The interactions with the robot lasted between five and eight minutes, where the robot informed the participants about the Ancient and Modern Wonders of the World. Besides providing information, the robot would ask the participants questions during the interactions. For example, it would ask participants whether they knew anything about the wonder that was being discussed. The interactions were pre-programmed to ensure interactions were similar for all participants.

### 3.2. Robot

The robot that was used for the experiment was the humanoid robot Pepper (approximately 1.5 m tall, developed by SoftBank Robotics [41]). An image of the robot is provided in Figure 1. We decided to use this robot as we wanted participants to focus on the behaviour the robot was displaying. As speech was included in the interactions, the use of a pet robot was not optimal. Larger humanoid robots were excluded as they may have appeared too mechanical or big and potentially intimidate participants. Smaller humanoid robots were excluded as they may have been perceived as toys. Furthermore, Pepper is a platform that has been used for human–robot interaction studies with older adults before [42–44]. It is also being introduced to the general public in settings such as shopping malls and museums [45,46], which may lead to future robots being inspired by the findings from the studies performed with Pepper. Therefore, the use of Pepper may make the findings of this work more applicable to future robots. The implementation of the expression of artificial emotions by Pepper was done using the software Choregraphe (version 2.5.5) [47]. This implementation of the emotion manipulation is described in earlier work [48].

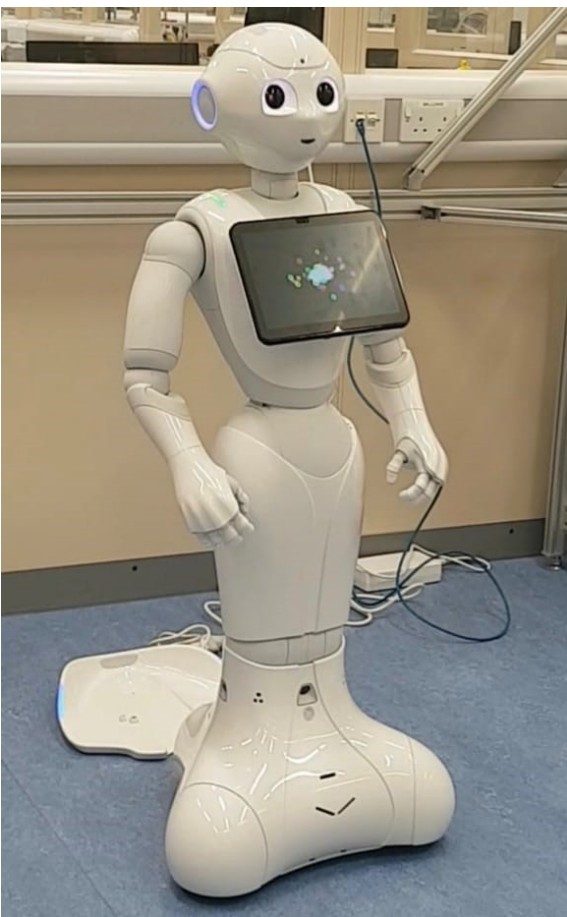

**Figure 1.** Social robot Pepper.

### *3.3. Measures*

There are no widely used methods to measure emotional deception and human–robot attachment. Furthermore, it is difficult to measure user experiences and potential consequences during human–robot interactions. Therefore, a variety of measurements were used in this study. Questionnaires were used to determine whether emotional deception occurred and whether participants became attached to the robot over time. Video recordings were taken for behaviour analysis to determine whether participants showed signs of discomfort during the interactions. This would allow identifying what robot behaviours would cause discomfort. Audio recordings were taken to determine whether participants' arousal changed due to the behaviour the robot was displaying. An increase in arousal can either follow a positive or a negative experience. If the latter, it may be an indicator of stress and highlight a negative impact of the human–robot interaction, raising ethical concerns. Arousal has already successfully been recorded in a human–robot interaction experiment, where it was found that arousal increased when participants witnessed a robot being tortured while it remained the same when witnessing a pleasant situation [49]. Finally, physiology sensors were also used to determine whether arousal levels changed depending on the robot's displayed behaviour.

### 3.3.1. Questionnaires

The aim of this work is to establish whether HRI experiments can be used to investigate concerns arising in the field of robot ethics. Therefore, a brief description of the questionnaires will be provided here for the purpose of explaining the experiment. The aim was to use well-established questionnaires in HRI for this experiment, as the use of such measurements would allow us to determine whether they could be used to investigate robot ethics without needing to consider the validity of the measurements. We used the

Almere model of trust [50] as this questionnaire allowed us to determine to what extent participants perceived the robot as a social being, also called 'social presence' (e.g., 'I can imagine the robot to be a living creature' and 'Sometimes the robot seems to have real feelings'). We also used the Godspeed questionnaire [51]. Through this questionnaire, we were able to determine whether participants attributed human traits to non-human entities (also known as 'anthropomorphism'), as participants had to indicate to what extent they perceived the robot's behaviour as 'lifelike' or 'conscious'. We compared participants' scores for these questionnaires when the robot expressed artificial emotions with their scores when it did not, which allowed us to determine whether participants were potentially deceived by the robot's expression of artificial emotion.

At the time of writing, there are no established questionnaires that investigate human–robot attachment. Inspiration can be taken from human–human attachment, human–object attachment, or human-pet attachment. Due to the nature of the interactions of the experiment, it was decided that object–attachment questionnaires were best suited for this study. Therefore, a questionnaire for object attachment [52] was adapted where the phrase 'the product' was replaced by the name of the robot (e.g., 'I have a bond with this product' was changed to 'I have a bond with Pepper'). Furthermore, there was a final interview after the experiment was completed where participants were asked whether they would miss the robot and whether they would want to use it in the future, which could indicate an increase in attachment feelings.

Finally, participants' mood was measured on several occasions during the experiment to determine whether their mood was impacted by the interactions with the robot. This was done measuring positive and negative affect, both explicit [53] and implicit [54]. For the explicit affect questionnaire, participants had to indicate to what extent several positive and negative emotions applied to them in that moment. For the implicit affect questionnaire, participants were given a list of non-existing words and were asked whether these words sounded positive or negative to them. These questionnaires helped indicate whether participants were feeling positive or negative at the time.

A more detailed description of these questionnaires has been provided in earlier work [31].

### 3.3.2. Video Recordings

Video recordings were taken to analyse participants' behaviour. We decided to include this measurement based on work indicating that nonverbal behaviour may be more reflective of impressions than speech [55]. Several observations over an extended period of time were necessary to be able to understand user experience during human–robot interactions [56]. All interactions were video-recorded from the point of view of the robot, with the participant central in the frame, and from the side, where both the robot and participant were visible. Three traditional coding principles for non-verbal behaviour are generic, restrictive, and evaluative coding [55,57]. Generic and restrictive coding focus on behavioural and postural descriptions such as body positions and hand gestures, whereas evaluative coding focuses on subjective impressions from the observers. It was decided to use a combination of generic and restrictive coding as the experience from the coders in this project was too limited to allow for evaluative coding. The camera that was used to record participants from the front is a Nikon D5100. The side view was recorded using a Samsung Galaxy S7.

### 3.3.3. Audio Recordings

Audio recordings allowed for the analysis of speech prosody data, which can be an indicator of increased arousal, both positive and negative [58]. More directly, it was determined that speech prosody can be an indicator of stress [59]. Features that were investigated in this study are pitch (min, max, mean, and standard deviation) and intensity (min, max, mean, and standard deviation). Together with duration, these are the three

features most often used in speech prosody analysis [60]. However, duration was not included in this project due to the nature of the interaction.

### 3.3.4. Physiology

Physiological data can also indicate an increase in arousal. In this project, we recorded heart rate variability (HRV) and electrodermal activity (EDA), as they are both indicators of increased arousal and potentially stress [61]. HRV has been used in HRI experiments before, where a difference in stress level was found based on the trajectory that a robot used to approach participants [62]. EDA has been used in HRI experiments as well, where it was found that EDA differed when participants witnessed specific robot motions [63]. The sensor used to gather these data was a Shimmer GSR+ sensor (http://www.shimmersensing.com/products/shimmer3-wireless-gsr-sensor, accessed on 20 September 2021). One sensor to measure HRV was placed on participants' earlobe, and two sensors were placed at the base of the middle- and ring-finger of participants' non-dominant hand to measure EDA.

### *3.4. Participants*

Fourteen participants (5 female, 9 male, age $M = 76.3$, $SD = 8.5$) completed the experiment. Four participants interacted only with the robot that did not express artificial emotions. Ten participants interacted with the robot that sometimes expressed artificial emotions and sometimes did not.

## 4. Results

### *4.1. Questionnaires*

The results from the questionnaires have been discussed in detail in earlier work [31]. A brief summary for results on emotional deception, emotional attachment, and participants' mood can be found below.

### 4.1.1. Emotional Deception

Items measuring emotional deception were anthropomorphism and social presence. The robot's behaviour did not significantly impact ratings of anthropomorphism, indicating that the robot that displayed emotions was not anthropomorphised more or less than the robot that did not display emotions, nor were there any changes over time. Social presence was rated significantly higher by participants who interacted with the robot that displayed emotions, compared to participants who only interacted with the robot that did not display emotions.

### 4.1.2. Emotional Attachment

It was found that attachment was low for most participants and that it did not change over time. For some participants, attachment was high and remained high for the duration of the experiment. Participants' level of attachment significantly impacted their perception of the robot as a social being, but did not impact their tendency to anthropomorphise the robot, as shown in Figures 2 and 3. It was found that not only participants who were highly attached to the robot, but also some participants who were marginally attached to the robot, responded 'yes' to the question whether they would miss the robot. Most participants indicated they would like to use the robot in the future.

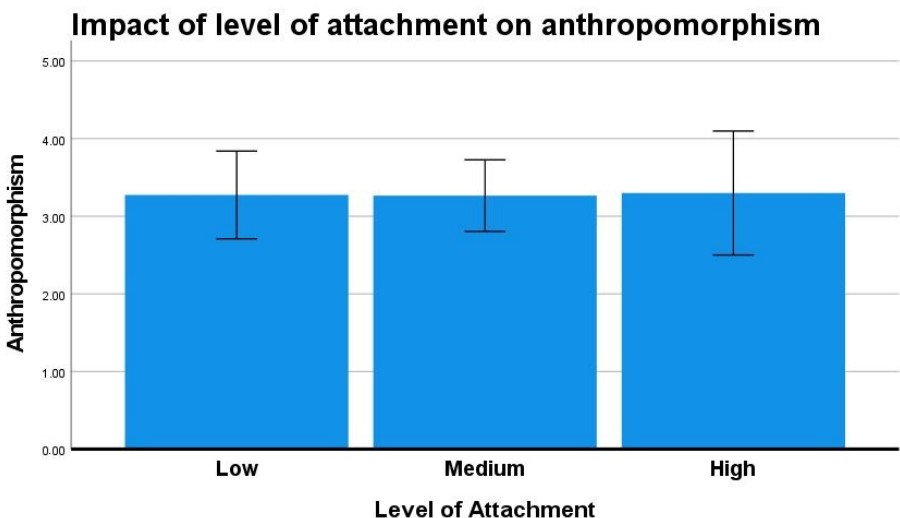

**Figure 2.** Participants' tendency to anthropomorphise the robot from not at all (1) to very much (5), based on their level of attachment to the robot (low, medium, high).

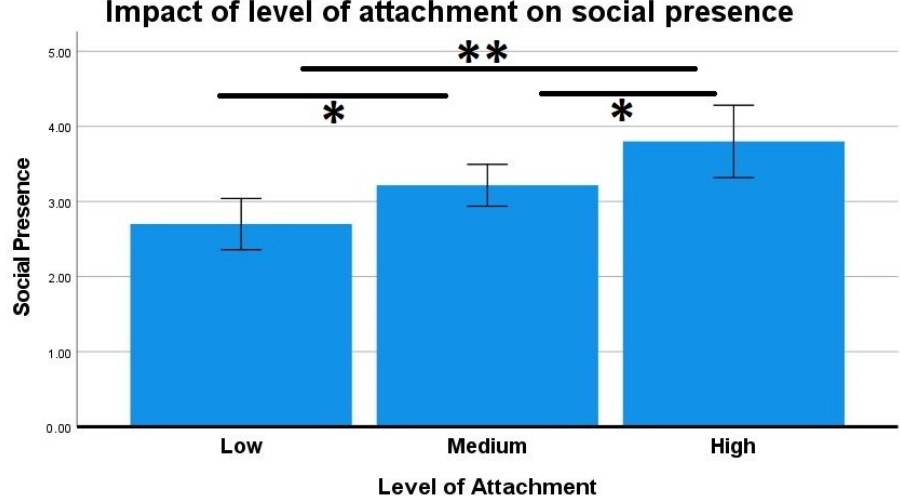

**Figure 3.** * $p < 0.05$, ** $p < 0.01$; Participants' perception of the robot as a social entity from not at all (1) to very much (5), based on their level of attachment to the robot (low, medium, high).

### 4.1.3. Mood

Participants' positive and negative affect was measured several times to determine whether the robot's behaviour impacted their mood. Both an explicit and an implicit questionnaire were used for this. The explicit questionnaire indicated that negative affect significantly decreased over time. This was expected, as negative affect represents feelings of unease, and it was expected that these feelings would decrease once participants became familiar with the robot. The implicit questionnaire was not used for analysis due to difficulty with completion.

### 4.2. Video Recordings

While coding the video-recordings, it was found that participants often displayed behaviours based on the content of the interaction and not the behaviour the robot was displaying. For example, when the robot was informing participants on the Great Wall of China, it would ask participants whether they could guess the total length of it. If it turned out their guess was off, they would laugh. Even though this makes for very interesting behaviour analysis, it was not suitable to determine whether the behaviour displayed by participants was a result of the display of emotions by the robot. Therefore, the data

gathered through the video recordings could not be used to analyse the impact of displayed robot behaviour.

### 4.3. Audio-Recordings

Audio recordings were extracted from the video recordings and transformed into audio files. The sections where participants were speaking in these audio files were cut, and these fragments were analysed using the software Praat (Version 6.1.03) [64], a software program that is often used in speech pathology studies [58]. These fragments could only be compared within participants, as pitch and intensity levels differ per person. Paired samples *t*-tests indicated that there were some differences based on the robot's behaviour. For example, the mean pitch was significantly higher for one participant when they interacted with the robot that displayed emotions compared to when they interacted with the robot that did not display emotions. Furthermore, the maximum intensity used to talk to the robot that displayed emotions was significantly lower for one participant compared to when they interacted with the robot that did not display emotions. However, as these findings were not consistent for all participants, no conclusions could be drawn from these results.

### 4.4. Physiology

#### 4.4.1. Heart Rate Variability

HRV was measured through photoplethysmography (PPG)—a technique that measures blood volume. There are several measurements that describe HRV. The one used in this project was RMSSD—the root mean square of successive differences, which reflects the differences between heart beats and is often used for short-term nature of the interactions [65,66], such as the interactions from this experiment. The software used to analyse these data are Kubios [67]. It provides RMSSD values for each participant for each interaction with the robot. The Kubios user guide provides ranges for RMSSD values that go from 'very low' to 'high', and nearly all acquired values fell within the 'normal' range. No difference was found based on the displayed robot behaviour.

#### 4.4.2. Electrodermal Activity

The EDA data gathered turned out to be unreliable, as values ranged from 0.02 µs to 0.07 µs, where reliable values range from 1 to 20 µs. It was not a malfunction of the device, as testing it on ourselves and colleagues provided reliable values. Nonetheless, the EDA data could not be used for analysis.

### 4.5. Incidental Findings

At the start of the experiments, it was found that participants had issues understanding what the robot was saying. This was not found when pilot testing the experiment. The volume of the robot's speech was sufficient as participants were able to hear the robot, but they were not able to understand its articulation. This was resolved by providing subtitles that were displayed on the tablet located on Pepper's chest. This incidental finding has been discussed in earlier work [68], and it does not focus on emotional deception or emotional attachment. However, due to its important ethical implications, especially as participants put blame on themselves and not the technology, it is included in this work as well. It will be discussed in more detail in the following section.

## 5. Discussion

A longitudinal human–robot interaction study was conducted to determine the potential impact of expression of artificial emotions by social robots on self-reported healthy older adults. A range of measurements were used to determine a potential occurrence of emotional deception and emotional attachment during these interactions. The use of some measurements was more successful than others, which will be discussed in the following sections.

*5.1. Questionnaires*

The conclusion that can be drawn from the results of the questionnaires is that the expression of artificial emotion may have led to emotional deception and emotional attachment for some participants and that there may be a relationship between the two. This conclusion highlights that more research is needed to better understand the user experience when a social robot expresses artificial emotions.

It should be noted that the questionnaires that were used are well established in HRI research but may not have been suitable for this experiment. Emotional deception was measured through anthropomorphism and social presence, as high levels of either or both may indicate an incorrect understanding of the robot's internal state. It is likely that other questionnaires on these topics, perhaps less well-known in HRI research, may have resulted in different and perhaps more insightful outcomes. This indicates that an empirical approach to robot ethics is useful but should be extended beyond HRI to include social sciences where there is more experience in measuring attitudes and perceptions. However, it is also possible that the questionnaires used are indeed suitable and that the expression of artificial emotion did not impact participants' user experience. This will need to be determined in future work.

Emotional attachment was measured by adapting a consumer-product attachment questionnaire and asking participants whether they would miss the robot and would want to use it in the future. Findings indicate that even though attachment was low for most participants, several participants indicated they would miss the robot, and most participants would like to use the robot in the future. This indicates that adapting the consumer-product attachment questionnaire was not sufficient to measure human–robot attachment and a new approach should be considered for future work. Furthermore, it also appears that even though attachment to a robot may lead to increased use [9], it does not necessarily mean that willingness to use is an indicator for attachment. These findings are supported by the argument that social robots should be identified as a new ontological class [56,69], which means a new questionnaire specifically for human–robot attachment should be developed to best understand the user experience.

*5.2. Other Measurements*

The other measurements used in this experiment (video/audio-recordings, physiology) provided multiple challenges, either because the data gathered were not appropriate for analysis or because the data did not provide any significant insights. However, it should be noted that the emotion manipulation in this experiment was designed to be low, so it is possible that the sensors used were not able to detect the impact.

The video recordings provided interesting data on user experience, but the behaviour participants displayed did not necessarily follow from the behaviour the robot was displaying. Therefore, it was not possible to draw conclusions regarding potential negative consequences of expression of artificial emotion from the video recordings. Analysis of the video recordings indicated that participants displayed behaviours following several triggers and not just the robot's displayed behaviour. This suggests that behaviour analysis through video recordings is not suitable for analysing the impact of a specific robot feature.

The audio recordings did provide data that could be analysed. However, interpersonal differences may have impacted the results, meaning no strong conclusions could be drawn, as significant findings were not consistent for all participants. Furthermore, it is possible that the number of data gathered through these audio-recordings was not large enough to draw conclusions, as the input from participants was limited due to the nature of the interaction.

Finally, physiological data were gathered to support other behavioural measures. However, the data gathered in this study did not support any other measures. EDA data was not suitable for analysis, and HRV data did not indicate any differences based on the robot's behaviour. It should however be noted that the emotion manipulation by

the robot was minimal and therefore the sensor may not have been able to pick up on these differences.

The results—or lack thereof—from these technologies indicate that the technologies are not yet advanced enough to be used in all human–robot interactions and to investigate robot ethics. This raises questions regarding HRI research that is currently being performed—are we asking the right research questions and should we even continue research in this area if we cannot sufficiently measure potential consequences of current developments? Should we shift the focus from developing social robots to developing methods to understand the impact—both intended and unintended—of social robots? Perhaps the effort to develop responsible social robots with minimal negative impact of human–robot interactions on individuals and society should be increased?

### 5.3. Insights from Incidental Findings

It was found that participants had trouble understanding the robot, supporting the concern that the use of new technologies can lead to accessibility issues. Such issues are more likely for vulnerable populations, and even more where ethical concerns with potential psychological impact are being investigated. Research and new technologies should be inclusive, and a potential limitation of inclusivity should be considered given this finding. Not considering these aspects of HRI may increase the divide of opportunities for people in specific populations. This was highlighted by the fact that participants had issues understanding the robot, and that the physiology sensor used did not gather appropriate EDA data from participants, whereas this was not a problem when piloting the study with colleagues.

Finally, researchers in this area should acknowledge the psychological impact that these experiments may have on participants and that they, especially when considered vulnerable, can become agitated. This may occur even when precautions have been taken and protocol to protect the participant has been followed. Risk assessments should include this concern, not only to protect the participant from potential harm but also to protect those conducting the research when unanticipated negative events occur.

### 5.4. Limitations of the Study

The experiment had a mixed between–within subject design. This was necessary as it was expected that the number of participants would be low, and this would allow us to ivestigate the impact of expression of artificial emotions by a robot both between and within subjects. Investigating both aspects resulted in a more complete understanding of the user experience.

The emotion manipulation by the robot was designed to be low, which meant that the differences between the robot expressing an artificial emotion or not were subtle. This may have led to a lack of response or a low level response, whereas differences may have been detected if the differences had been more extreme. However, due to the novelty of the approach to using HRI to investigate robot ethics, and additional ethical concerns if the emotion manipulation were more pronounced, it was decided to keep the manipulation low for this study.

Finally, one could argue that the target group in this work may be considered too small to draw conclusions for the wider field of robot ethics. However, older adults are reflective of a population at large with different generational experiences, and we might see perspectives change over time. Considering this, we believe it is better to address small target groups when investigating robot ethics to be able to better understand the user experience for that target group. This research can then be extended to include other target groups in future work, such as mixed societies [70], and different physical human–robot interactions such as support from exoskeletons [3] and industrial settings [71].

## 6. Conclusions

The goal of this research was to determine whether robot ethics can be investigated through human–robot interactions. One reason for doing so is that it allows us to address ethical concerns that are raised in the literature, which improves our understanding of potential negative consequences. Furthermore, identifying ethical concerns in the literature may restrain the development of substandard robots and lead to the development of new technologies that incorporate an ethical framework.

It was found that the means to perform such research is limited, and not inclusive for all target groups. However, results also indicate that it is possible to gather initial insights on ethical concerns through HRI studies. The insights from this project raise questions regarding current social robot research, and whether the focus of this research should be extended to include the development of a variety of HRI measurements. Incorporating these insights into studies will improve our understanding of user experience during human–robot interactions, allowing us to research ethical concerns of social robots and ensuring they can safely be deployed in the world.

**Author Contributions:** Conceptualization, A.v.M., N.Z., S.D., M.S., A.W. and P.C.-S.; methodology, A.v.M., N.Z. and P.C.-S.; software, A.v.M.; validation, A.v.M., N.Z. and P.C.-S.; formal analysis, A.v.M. and N.Z.; investigation, A.v.M.; resources, N.Z. and P.C.-S.; data curation, A.v.M.; writing—original draft preparation, A.v.M.; writing—review and editing, A.v.M., N.Z., S.D., M.S., A.W. and P.C.-S.; visualization, A.v.M.; supervision, N.Z., S.D., M.S., A.W. and P.C.-S.; project administration, A.v.M. and P.C.-S.; funding acquisition, N.Z., S.D., M.S., A.W. and P.C.-S. All authors have read and agreed to the published version of the manuscript.

**Funding:** This research and the APC were funded by the European Union's Horizon 2020 research and innovation programme under the Marie Skłodowska-Curie grant agreement No. 721619 for the SOCRATES project.

**Institutional Review Board Statement:** The study was conducted according to the guidelines of the Declaration of Helsinki and approved by the Ethics Committee of the University of the West of England. Protocol code: FET.18.02.030. Date of approval: 11 April 2018

**Informed Consent Statement:** Informed consent was obtained from all subjects involved in the study.

**Data Availability Statement:** The data presented in this study are not available due to the sensitive nature of the data, to protect participants' privacy.

**Acknowledgments:** The authors would like to thank the participants who volunteered to participate in this experiment.

**Conflicts of Interest:** The authors declare no conflict of interest.

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
