# Peer review of "A New Perspective on Robot Ethics through Investigating Human–Robot Interactions with Older Adults"

_applsci, doi:10.3390/app112110136_

Round 1
Reviewer 1 Report
The title and abstract are well chosen and well written. The authors have presented the work well.
Author Response
We would like to thank the reviewer for taking the time to read our work and for the positive feedback.
Reviewer 2 Report
The paper investigates an interesting topic. However, only a small target application area (social robots for the elderly) is considered.
Comments:
- would it be possible, at least in the introduction, to mention some other application areas in which ethics is important for human-robot collaboration? I guess the industrial sector could be mentioned. Some applications could be related to [1,2,3] ([1] for physical human-robot collaboration, [2] for preference-based optimization, [3] for exoskeleton devices);
- I would suggest creating a new Section where to put Sections 1.1 and 1.2 instead to have them in the introduction Section;
- a subsection for the paper contribution in the introduction Section would better highlight the aim of the paper;
- even if the questionnaire was already developed in another paper, it would be useful to describe it more in detail;
- it would be interesting to have a graphical analysis of the results;
- check the English.
[1] Roveda, Loris, et al. "Assisting operators in heavy industrial tasks: on the design of an optimized cooperative impedance fuzzy-controller with embedded safety rules." Frontiers in Robotics and AI 6 (2019): 75.
[2] Roveda, Loris, et al. "Pairwise Preferences-Based Optimization of a Path-Based Velocity Planner in Robotic Sealing Tasks." IEEE Robotics and Automation Letters 6.4 (2021): 6632-6639.
[3] Roveda, Loris, et al. "Design methodology of an active back-support exoskeleton with adaptable backbone-based kinematics." International Journal of Industrial Ergonomics 79 (2020): 102991.
Author Response
Thank you very much for your feedback. Please find our response to your comments below:
The paper investigates an interesting topic. However, only a small target application area (social robots for the elderly) is considered.
We would like to thank the reviewer for their comments. We have addressed them in more detail below. Regarding the small target area, we argue that this is necessary to fully understand the user experience and ensure the results include experiences from all participants within a certain target area. As supported by our results, even for the small group of older adults it was already difficult to ensure inclusivity to highlight all possible concerns. Therefore, in order to investigate robot ethics through human-robot interactions, we believe it is necessary to focus on small target application areas. We have discussed this in more detail in line 407-412. We have also adapted to title and it now includes the target application area so readers know what to expect.
1. would it be possible, at least in the introduction, to mention some other application areas in which ethics is important for human-robot collaboration? I guess the industrial sector could be mentioned. Some applications could be related to [1,2,3] ([1] for physical human-robot collaboration, [2] for preference-based optimization, [3] for exoskeleton devices);
Thank you for this suggestion, we have updated the introduction and it now mentions other applications where ethics is important. We also mention these other areas as options for future work (line 412-414).
2. I would suggest creating a new Section where to put Sections 1.1 and 1.2 instead to have them in the introduction Section;
We understand the reviewer’s point of view. We originally included this in the introduction to adhere to the suggested manuscript outlined in the instructions for authors. However, we agree with the reviewer that these sections should not be part of the introduction and have placed them in a new section called ‘Emotional Deception and Emotional Attachment’.
3. a subsection for the paper contribution in the introduction Section would better highlight the aim of the paper;
Thank you for this comment, we have added this to the introduction (line 56-60).
4. even if the questionnaire was already developed in another paper, it would be useful to describe it more in detail;
We used several existing questionnaires that have been established in HRI research, we did not develop them ourselves. This has been explained more clearly now in line 152-154. To give the reader a more clear idea of what kind of questions were asked, we have added some example questions to the description.
5. it would be interesting to have a graphical analysis of the results;
We have added a graphical analysis of the results.
6. check the English.
Thank you for this comment, we have gone through the manuscript and improved the English where needed.
Reviewer 3 Report
Authors completely missed the several aspects of organism-robot interaction, excpecially animal-robot interaction, mixed societies, and ethorobotics, that is at the base of social robotics and ethics. Indeed, humans are animals, and animal models are ogten used to investigate behavioural processes and neuroethological isssues that are related to humans.
Authors should inlcude and discuss (maybe in a dedicated section) these important contexts.
Some relevant examples are
Romano, D., & Stefanini, C. (2021). Unveiling social distancing mechanisms via a fish-robot hybrid interaction. Biological Cybernetics, 1-9.
Ladu, F., Mwaffo, V., Li, J., Macrì, S., & Porfiri, M. (2015). Acute caffeine administration affects zebrafish response to a robotic stimulus. Behavioural Brain Research, 289, 48-54.
Auhtors may find many additional recent examples (including for instance the use of robot and animal models for social information, lateralization studies, etc) and review articles to improve the scientific value of their work that now is very weak and narrow to be considered for acceptance.
A deep English revision is needed.
Author Response
We appreciate the comments from the reviewer and understand their point of view. The reviewer has highlighted a key aspect of the growing field of social robotics and its many applications due to the diverse range of social robot platforms. We have mentioned the wider application area of social robotics and ethics in the introduction, and have now added a paragraph in the limitations section (section 6.4) where we present why we investigated a small target application area. We have also updated the title to include this. In the discussion of the paper, we refer back to the suggestions of the reviewer and mention these as future work (line 412-414). Finally, we have gone through the manuscript and revised the English.
Round 2
Reviewer 2 Report
The paper can now be accepted.
Author Response
We thank the reviewer for their feedback and in helping us improve the quality of our manuscript.
Reviewer 3 Report
Authors addressed almost completely my comments. Just minor issues related to the text clarity should be considered. in particular, there are sevral parts difficult to follow . I suggest Authors to read carefully their work and polishing the text to increase its readability.
Also auhtor may add the following references to the sentence "Interactions with robots reach beyond the scope of human-robot interactions, for example animal-robot interactions...".
Romano, D., Benelli, G., Hwang, J. S., & Stefanini, C. (2019). Fighting fish love robots: mate discrimination in males of a highly territorial fish by using female-mimicking robotic cues. Hydrobiologia, 833(1), 185-196.
Yang, Y., Clément, R. J., Ghirlanda, S., & Porfiri, M. (2019). A comparison of individual learning and social learning in zebrafish through an ethorobotics approach. Frontiers in Robotics and AI, 6, 71.
Author Response
Authors addressed almost completely my comments. Just minor issues related to the text clarity should be considered. in particular, there are sevral parts difficult to follow . I suggest Authors to read carefully their work and polishing the text to increase its readability.
We thank the reviewer for their continued feedback. We have thoroughly read through the manuscript again, and updated sentences that were quite long, and corrected sentences that were grammatically incorrect.
Also auhtor may add the following references to the sentence "Interactions with robots reach beyond the scope of human-robot interactions, for example animal-robot interactions...".
We appreciate the references suggested by the reviewer and have added them to this sentence.